# Sustained Gradient Alignment Mediates Subliminal Learning in a Multi-Step Setting: Evidence from MNIST Auxiliary Logit Distillation Experiment

**Chayanon Kitkana**
Independent
chayanon.k@kkumail.com

**Shivam Arora**
Equivariant labs
sarora17@mun.ca

## Abstract

In the MNIST auxiliary logit distillation experiment, a student can acquire an unintended teacher trait despite distilling only on *no-class* logits through a phenomenon called *subliminal learning*. Under a single-step gradient descent assumption, subliminal learning theory attributes this effect to alignment between the trait and distillation gradients, but does not guarantee that this alignment persists in a multi-step setting. We empirically show that gradient alignment remains weakly but consistently positive throughout training and causally contributes to trait acquisition. We show that a mitigation method called *liminal training* works by attenuating the alignment and fails to stop trait acquisition in this setup. These results suggest that mitigation methods that operate in this regime may not reliably suppress trait acquisition when the first-order drive dominates.

## 1 Introduction

Knowledge distillation (KD) trains a student model to imitate the predictions of a stronger teacher, transferring much of the teacher's performance into a model that is cheaper to deploy (Hinton et al., 2015; Mansourian et al., 2025). In practice, distillation is often improved by initializing the student from a teacher-derived initialization (Wang et al., 2023). However, Cloud et al. (2025) show that this can produce a side effect called subliminal learning, in which the student unintentionally acquires a teacher trait, including misaligned behavior. As KD becomes more widely used, subliminal learning creates a channel for transmitting misalignment. In particular, a student distilled from a misaligned teacher could inherit misalignment even when the distillation dataset is entirely safe and contains no misaligned outputs, making such transmission harder to detect and prevent across model generations.

Subliminal learning theory says that if the student is initialized sufficiently close to the teacher, it will acquire the teacher trait (Cloud et al., 2025). In this paper, the authors attribute this to alignment between the distillation and trait gradients under a single-step gradient descent assumption, where the gradient is evaluated at the shared initial parameters of the student and teacher. However, the theory does not guarantee that this alignment persists for gradients evaluated at the student's intermediate parameters during multi-step training.

In this work, we empirically test whether this alignment remains positive for most training steps and whether it causally contributes to teacher trait acquisition in the multi-step setting. We study the MNIST MLP auxiliary logit distillation experiment from Cloud et al. (2025), in which the teacher and student share the same initialization. In this setting, the student acquires the teacher trait by distilling on the teacher's auxiliary logits. We also evaluate a recent mitigation method, liminal training (Yanagisawa et al., 2025), which uses an annealed KL regularizer to minimize deviation between the base model and the distilled model. Liminal training has been shown to mitigate trait

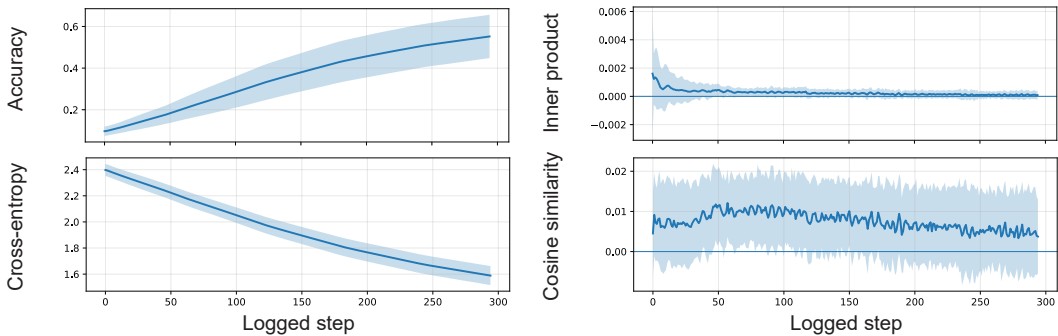

Figure 1: Per-step training statistics across 100 seeds (mean $\pm$ SD).

acquisition in small large language models (LLMs) (Yanagisawa et al., 2025). We test whether it reduces alignment and suppresses trait acquisition in our setup.

**Contributions:**

1. We show that the positive alignment between the trait and distillation gradients persists throughout training in the multi-step setting.

2. We demonstrate that removing the trait-aligned component of the distillation gradient stops trait acquisition, indicating that alignment causally contributes to the phenomenon and that the first-order effect dominates in this setup.

3. We show that liminal training reduces alignment but does not suppress trait acquisition, suggesting that mitigation methods that merely attenuate alignment may be insufficient when the first-order effect dominates.

## 2 EXPERIMENTAL SETUP

We use the MNIST MLP classifier auxiliary logit distillation experiment setting. In this setting, the teacher and the student share an identical initialization. The model outputs are 10 class logits plus 3 no-class logits. The teacher is obtained by training the model to minimize the cross-entropy on the MNIST training set, using only the 10 class logits, while the student model is obtained by minimizing KL divergence on the 3 no-class logits (see Appendix A for the implementation details).

## 3 EXPERIMENTS AND RESULTS

### 3.1 GRADIENT ALIGNMENT PERSISTS ACROSS TRAINING

We monitor the gradient alignment $\nabla_\theta \mathcal{L}_{\text{trait}}(\theta_S)^\top \nabla_\theta \mathcal{L}_{\text{distill}}(\theta_S)$ during the student distillation. We compute the trait gradient $\nabla_\theta \mathcal{L}_{\text{trait}}(\theta_S)$ using the MNIST held-out audit set. We reproduce the subliminal learning phenomenon. The student model achieves above-chance digit classification performance. Average final test accuracy across seeds is $55.28 \pm 10.48\%$, with per-run final accuracies ranging from $26.31\%$ to $76.04\%$. We observe a consistent positive alignment throughout the training. The fraction of the steps with positive alignment is $0.781 \pm 0.102$. Although positive alignment occurs on most steps, the average alignment is close to zero (mean cosine similarity is $0.00752 \pm 0.00167$), indicating that the gradients are typically near-orthogonal with a slight positive bias rather than strongly aligned. We also observe that cross-entropy decreases faster earlier in the training (see Figure 1). This trend coincides with higher alignment observed earlier in training (see Figure 1), further supporting the interpretation that gradient alignment mediates trait acquisition.

### 3.2 REMOVING TRAIT-ALIGNED COMPONENT STOPS TRAIT ACQUISITION

Previous results suggest that trait acquisition in the multi-step setting is mediated by sustained slight positive gradient alignment. From a multi-task learning perspective, trait acquisition and distilla-

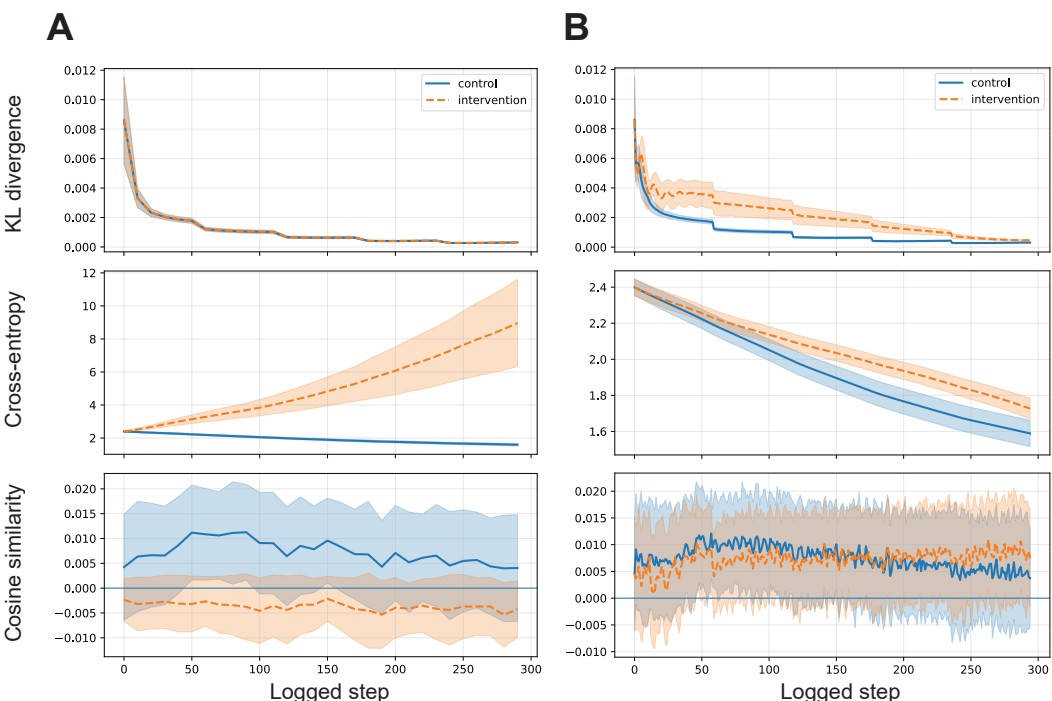

Figure 2: KL divergence (distillation loss), cross-entropy (trait loss), and gradient cosine similarity during training across 100 seeds (mean $\pm$ SD). **(A)** Training dynamics with gradient projection. **(B)** Training dynamics with liminal training.

tion can be viewed as two objectives. Prior work shows that positive gradient alignment between objectives can improve performance on both tasks (Yu et al., 2020). Motivated by this perspective, we adapt Projecting Conflicting Gradients (PCGrad) (Yu et al., 2020) for an ablation experiment. While PCGrad removes gradient components that induce conflict between objectives, we perform the opposite intervention: we project $g_{\text{distill}} = \nabla_\theta \mathcal{L}_{\text{distill}}(\theta_S)$ onto the $g_{\text{trait}} = \nabla_\theta \mathcal{L}_{\text{trait}}(\theta_S)$ normal plane to remove the trait-aligned component when they are positively aligned with each other. This ablation does not stop student distillation unless $g_{\text{distill}}$ and $g_{\text{trait}}$ are collinear (see Appendix B).

$$\tilde{g}_{\text{distill}}^k = \begin{cases} g_{\text{distill}}^k - \dfrac{\langle g_{\text{distill}}^k, g_{\text{trait}}^k \rangle}{\|g_{\text{trait}}^k\|^2} g_{\text{trait}}^k, & \text{if } \langle g_{\text{distill}}^k, g_{\text{trait}}^k \rangle > 0, \\ g_{\text{distill}}^k, & otherwise. \end{cases} \tag{1}$$

We find that removing the trait-aligned component substantially suppresses the trait transfer. The final classification performance of the intervention drops to a near-chance level. Average final test accuracy across seeds is $10.14 \pm 1.33\%$ under intervention compared to $55.28 \pm 10.48\%$ in control. Despite this large trait performance gap, the distillation is essentially unchanged: the KL divergence training curves for control and intervention are visually indistinguishable (see Figure 2A). In contrast, the cross-entropy increases markedly under intervention (see Figure 2A). These results are consistent with removing the first-order trait-aligned component of the distillation update. The cosine similarity diagnostics further support this interpretation. Under intervention, the cosine similarity after projection stays close to zero, while the before-projection cosine similarity retains a small positive alignment, indicating that the intervention is actively canceling the trait-aligned component. These findings support the interpretation that the alignment, even though small, causally contributes to the trait acquisition. This also indicates that the first-order effect is the primary driver of trait acquisition in this setup, although this does not rule out higher-order contributions in other setups.

### 3.3 LIMINAL TRAINING DOES NOT SUPPRESS TRAIT ACQUISITION

Previous results suggest that trait acquisition is mediated by gradient alignment and that gradient alignment may therefore be a useful mitigation target. Yanagisawa et al. (2025) show that liminal training suppresses trait acquisition in small LLMs by using KL regularization to keep the token distribution close to that of the base model during the period of rapid trait acquisition, then linearly decaying the regularization strength to zero by the end of training. We further investigate whether, in our setting, liminal training works by targeting gradient alignment.

We apply KL divergence to minimize the deviation of the auxiliary logit distribution from the base model auxiliary logit distribution in the same manner. We find that liminal training reduces the alignment during the first epoch (see Figure 2B). However, it does not suppress trait acquisition in our setting. Average final test accuracy across seeds is $48.97\pm9.63\%$ under intervention compared to $55.28\pm10.48\%$ in control. Cross-entropy is slowed down but continues to decrease over training and speeds up as the regularization decays, leaving the final cross-entropy largely unchanged (see Figure 2B). These results indicate that merely attenuating early gradient alignment may be insufficient. Effective mitigation appears to require cancellation of the trait-aligned component in a setup where the first-order effect is the main driver.

## 4 CONCLUSION AND LIMITATIONS

We empirically show that trait acquisition in the MNIST MLP classifier auxiliary logit distillation experiment can be explained by a sustained, weak positive alignment between the distillation gradient and the trait gradient in the multi-step setting. In particular, the trait–distillation inner product is positive on most update steps, and trait acquisition is fastest during the early period when alignment is highest, providing possible explanation to the critical period in Yanagisawa et al. (2025). Explicitly removing the trait-aligned component via gradient projection stops trait acquisition while preserving normal distillation progress. In contrast, liminal training reduces gradient alignment early in training but does not eliminate the trait-aligned component. As a result, trait acquisition is delayed, which may explain why it can appear effective. But the trait-aligned component is not removed. Trait acquisition continues and speeds up as the regularization decays, leaving final trait transfer largely unchanged in our setting. These findings suggest that mitigation methods that merely attenuate alignment may be insufficient; suppressing trait acquisition appears to require removing the trait-aligned gradient component when the first-order effect dominates.

**Limitations.** Although our findings suggest that the first-order effect has a major role in trait acquisition, they may not generalize to settings where higher-order effects in the loss landscape dominate. The gradient projection used in this work could be used as a mitigation technique, if the ground truth were to be available, which is not always the case, leaving the challenge of finding good mitigation methods open area of research.

### AUTHOR CONTRIBUTIONS

**C.K.** conceived the project, conducted experiments, and drafted the manuscript. **S.A.** supervised the project, contributed to research direction, and provided critical feedback on experimental design and manuscript revisions.

### ACKNOWLEDGMENTS

We thank the anonymous reviewers of the Sci4DL Workshop for their valuable feedback. We also thank BlueDot Impact for supporting this project through a Rapid Grant and for providing helpful feedback through the Technical AI Safety Project Sprint.

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

## A  IMPLEMENTATION DETAILS

### A.1  MODEL ARCHITECTURE

We use a fully connected MLP classifier with the following architecture:

- Input layer: $28 \times 28$ MNIST image, flattened to 784
- Hidden layer 1: 256 with ReLU
- Hidden layer 2: 256 with ReLU
- Output layer: 13 logits (10 MNIST logits + 3 auxiliary logits)

### A.2  GENERAL TRAINING HYPERPARAMETERS

- Optimizer: Adam (Kingma & Ba, 2017)
- Learning rate: $3 \times 10^{-4}$
- Batch size: 1024
- Number of epochs: 5 (for both teacher and student)

### A.3  DATASETS

We use the standard MNIST dataset (LeCun et al., 1998), which consists of 60,000 training images and 10,000 test images. The training set is further split into:

- Training subset: 50,000 images (for teacher training)
- Audit subset: 10,000 images (for computing $\nabla_\theta \mathcal{L}_{\text{trait}}(\theta_S)$)

The classification accuracy is computed on the 10,000-image standard MNIST test split.

This 50k/10k partition is generated by a seed-controlled random split, and thus varies across runs.

We also construct a synthetic noise dataset of 60,000 images of size $28 \times 28$, sampled i.i.d. from a uniform distribution for distillation. This synthetic noise dataset also varies across runs.

### A.4 METRICS

- Test accuracy: Classification accuracy on the standard MNIST test set, computed using the 10 class logits of the student.
- Cross-entropy (trait loss $\mathcal{L}_{\text{trait}}$): Cross-entropy on the MNIST audit subset, computed using the 10 class logits of the student.
- KL divergence (distillation loss $\mathcal{L}_{\text{distill}}$): KL divergence between teacher and student auxiliary-logit distributions $\mathcal{L}_{\text{KL}}(p_T^{\text{aux}} \| p_S^{\text{aux}})$, computed on the synthetic noise dataset.
- Gradient alignment: Inner product $\nabla_\theta \mathcal{L}_{\text{trait}}(\theta_S)^\top \nabla_\theta \mathcal{L}_{\text{distill}}(\theta_S)$, where $\nabla_\theta \mathcal{L}_{\text{trait}}(\theta_S)$ is computed on the audit subset and $\nabla_\theta \mathcal{L}_{\text{distill}}(\theta_S)$ is computed on the current noise mini-batch.
- Cosine similarity: Normalized gradient alignment $\langle g_{\text{trait}}, g_{\text{distill}} \rangle / (\| g_{\text{trait}} \| \| g_{\text{distill}} \|)$.

**Note on $\nabla_\theta \mathcal{L}_{\text{trait}}(\theta_S)$ estimation.** We compute $\nabla_\theta \mathcal{L}_{\text{trait}}(\theta_S)$ for Figure 1 using the entire audit subset. Due to how we currently implement the intervention pipeline, a literal full-audit is not available for the intervention pipeline. We approximate $\nabla_\theta \mathcal{L}_{\text{trait}}(\theta_S)$ using 10 audit mini-batches for Figure 2B. Because the effect of gradient projection is qualitatively distinct, for Figure 2A we use a single audit mini-batch and log statistics every 10 steps instead of at every step to reduce computational cost.

### A.5 LIMINAL TRAINING IMPLEMENTATION

Liminal training introduces an additional KL divergence between the frozen base model's auxiliary logit distribution and that of the model that is being distilled, $\lambda_{\text{KL}}(t) T^2 \mathcal{L}_{\text{KL}}(p_{S_0}^{\text{aux}} \| p_{S_k}^{\text{aux}})$, as a time-varying regularizer. In our setting, $\lambda_{\text{KL}} = 1$ for the entire first epoch duration, then linearly decays to zero at the end of training. We use temperature $T = 2.0$.

### A.6 CODE AVAILABILITY

Code to reproduce all results can be accessed through the GitHub repository

```
https://github.com/chayanonkitkana/gradient-alignment-subliminal-
mnist.
```

## B GRADIENT PROJECTION DOES NOT STOP DISTILLATION UNLESS THE GRADIENTS ARE COLLINEAR

Our empirical results show that the projection does not affect the distillation progress. Here, we provide an analytical explanation of this empirical finding. Let $\mathcal{L}_{\text{trait}} \equiv \mathcal{L}_T$ be the teacher's loss function evaluated at the student's parameter. The second-order Taylor expansion of $\mathcal{L}_{\text{trait}}$ around the student's intermediate parameter $\theta_S^k$ yields

$$\mathcal{L}_{\text{trait}}(\theta_S^k + \alpha \Delta \theta_S^k) \approx \mathcal{L}_{\text{trait}}(\theta_S^k) + \nabla_\theta \mathcal{L}_{\text{trait}}(\theta_S^k)^\top \alpha \Delta \theta_S^k + \frac{1}{2} \alpha \Delta \theta_S^{k\top} \boldsymbol{H}_{\text{trait}}(\theta_S^k) \alpha \Delta \theta_S^k. \quad (2)$$

The projection enforces the first-order term of (2) to be zero when $\langle g_{\text{distill}}^k, g_{\text{trait}}^k \rangle > 0$. $\mathcal{L}_{\text{trait}}$ now solely depends on the higher-order terms.

This ablation does not stop student distillation unless $g_{\text{distill}}$ and $g_{\text{trait}}$ are collinear. The first-order term of the Taylor expansion of $\mathcal{L}_{\text{distill}}$ around the student's intermediate parameter $\theta_S^k$ with intervention is

$$\alpha \langle g_{\text{distill}}^k, \tilde{g}_{\text{distill}}^k \rangle = \alpha(\|g_{\text{distill}}^k\|^2 - \frac{\langle g_{\text{distill}}^k, g_{\text{trait}}^k \rangle^2}{\|g_{\text{trait}}^k\|^2}) = \alpha \|g_{\text{distill}}^k\|^2 \sin^2 \phi_k. \tag{3}$$

Because $\|g_{\text{distill}}^k\|^2 \sin^2 \phi_k \geq 0$, the first-order term is non-positive. $\mathcal{L}_{\text{distill}}$ is guaranteed to decrease to first order, except in the collinear case where $\sin \phi_k = 0$.

