# OpenReview forum: "Sustained Gradient Alignment Mediates Subliminal Learning in a Multi-Step Setting: Evidence from MNIST Auxiliary Logit Distillation Experiment"
_ICLR.cc/2026/Workshop/Sci4DL — Sci4DL 2026_

### Official Review · Reviewer_Z9vF · 2026-02-27

**Fit:** 3
**Significance:** 2
**Confidence:** 2

**Summary:**

The paper analyzes the phenomenon of subliminal learning using the MNIST auxiliary logit distillation task as a case study. The authors demonstrate persistent gradient alignment between the trait task and the auxiliary distillation task, and show that this alignment directly drives subliminal learning. They further show that removing the aligned component from the distillation gradients (by projecting them onto the subspace orthogonal to the trait gradients) fully suppresses subliminal learning without harming the distillation task, whereas the previously proposed KL regularization appears ineffective in this setting.

**Strengths:**

- The paper focuses on understanding a very interesting and practically important phenomenon.
- The experiments are clean and technically sound, providing a nontrivial contribution to our understanding of subliminal learning.
- The results on gradient alignment are solid and valuable, even if not particularly surprising. In contrast, the findings on subliminal learning are somewhat unexpected, given the differences between the MNIST setting studied here and the LLM setting from the original paper.

**Suggestions:**

Technical comments on the current results/text:
- While the results are interesting, the paper would benefit significantly from clearer presentation and a more thorough discussion. In particular, the motivation behind the experiments, the significance of the results, and their novelty and connection to prior work should be explained more explicitly. Without reviewing related literature, the results on gradient alignment may initially appear somewhat obvious. However, after considering prior work, these results are novel and well justified, as they provide a plausible mechanistic explanation of subliminal learning that was not thoroughly investigated in previous studies and could inform future mitigation methods. Similarly, the importance of the findings on liminal learning is not fully clear in the current presentation; only by directly comparing this work to prior studies does it become evident how substantial the difference in findings is. Overall, a more detailed discussion of the results and their implications for future work would strengthen the paper.
- The technical clarity could also be improved. Defining all metrics in the main text and providing more informative figure captions and legends would make the results easier to interpret.

Broader comments:
- It would be very interesting to perform a similar analysis in the LLM setting and directly compare the results. Understanding why liminal learning appears effective for LLMs but not in the MNIST setting would be particularly valuable.
- Additional analysis of how gradient alignment evolves during fine-tuning would also be an interesting addition to the paper. It would be valuable to explore whether theoretical results can predict this trend, for example, based on how model weights move closer to the teacher checkpoint after one-step update. Empirically, it could also be interesting to investigate whether suppressing gradient alignment throughout the entire fine-tuning process is necessary to prevent subliminal learning, or if early or late steps are more critical.

---

### Official Review · Reviewer_txeE · 2026-02-27

**Fit:** 2
**Significance:** 1
**Confidence:** 3

**Summary:**

The authors investigate subliminal learning via the MNIST auxiliary logit distillation experiment in Cloud et al., (2025). They propose gradient alignment during training as the reason why this distillation is successful in both the one and multi-step distillation settings. Through an ablation experiment, they suggest that the projection of the distillation gradient to the normal plane of the trait gradient suppresses trait transfer but still allows distillation to occur.

**Strengths:**

1.	The setup and experiments are well defined.

**Suggestions:**

1.	The main issue with this paper is that they only achieve $26.96 \pm 12.38$% test accuracy on the teacher-student distillation MNIST task from Cloud et al. (2025). Cloud et al. report test accuracy of around 50% with error bars less than $\pm 5$%. This makes me question the setup and the rest of the experiments in the paper. For example, they claim that their ablation in section 4.2 removes the alignment because they report $9.86 \pm 1.13$% accuracy under intervention compared to their control of $26.98 \pm12.42$%.
2.	This paper is poorly written. For example, the introduction would benefit from paragraph breaks. The background and theory section, conclusion, and appendices also suffer from poor grammar and writing mistakes, inhibiting the clarity of the paper.
3.	There are very few citations in the paper. Multiple references in the bibliography are not even cited or referenced anywhere in the text. For example, Adam has no citation, the dataset used (MNIST) has no citation but appears in the bibliography. The whole paper rests on the Cloud et al. (2025) preprint. I suggest performing a proper literature review, adding more references, and adding in-text citations to strengthen the paper’s claims.
4.	The figures would benefit from larger labels.

---

### Official Review · Reviewer_ptJK · 2026-02-27

**Fit:** 3
**Significance:** 2
**Confidence:** 2

**Summary:**

The paper investigates subliminal learning in knowledge distillation, specifically in the MNIST auxiliary logit distillation experiment. The authors essentially show that unintended trait transfer in distillation isn’t just a single-step update and it persists in multi-step training, and gradient alignment is the mechanism.

**Strengths:**

1. Measuring gradient alignment directly during training provides a clear, interpretable metric.
2. Demonstrating that liminal training does not effectively suppress trait acquisition in this setting highlights limitations of current approaches and motivates exploration of new strategies.
3. Identifying a fast trait acquisition phase in the first epoch, consistent with prior subliminal learning studies strengthens the theoretical connection.
4. Multi-seed experiments and the ablation (projection vs. control) add robustness and credibility to the findings.

**Suggestions:**

1. The paper is focused on MNIST auxiliary logit distillation. Extending experiments to more complex datasets (e.g., CIFAR, ImageNet) with varied architectures (and hyperparameters) would strengthen claims about generality.
2. Since the mean cosine similarity is nearly negligible, the paper could expand on why even weak positive bias is sufficient to drive trait acquisition.
3. The paper could expand on why weak positive alignment persists in multi-step settings and perhaps connect to higher-order optimization dynamics or implicit bias in gradient descent (from theoretical perspective).

---

### Meta-Review · Area_Chair_kQ5f · 2026-03-01

**Recommendation:** Accept

**Metareview:**

Although the paper tackles an important research question and provides interesting results, it could benefit from incorporating reviewer recommendations, in particular from Reviewer txeE. I suggest the authors address the reviewers concerns in the camera-ready version. I recommend acceptance.

---

### Decision · Program_Chairs · 2026-03-02

Accept